# Indoxyl Sulfate Mediates the Low Inducibility of the NLRP3 Inflammasome in Hemodialysis Patients

**DOI:** 10.3390/toxins13010038

**Published:** 2021-01-07

**Authors:** Li-Chun Ho, Ting-Yun Wu, Tsun-Mei Lin, Hung-Hsiang Liou, Shih-Yuan Hung

**Affiliations:** 1Division of Nephrology, Department of Internal Medicine, E-DA Hospital, Kaohsiung 82445, Taiwan; hoim7024@ms3.hinet.net; 2Division of General Medicine, Department of Internal Medicine, E-DA Hospital, Kaohsiung 82445, Taiwan; 3School of Medicine, College of Medicine, I-Shou University, Kaohsiung 82445, Taiwan; 4Institute of Clinical Medicine, College of Medicine, National Cheng Kung University, Tainan 704017, Taiwan; b86316@hotmail.com; 5Department of Medical Laboratory Science, College of Medicine, I-Shou University, Kaohsiung 82445, Taiwan; ed100744@edah.org.tw; 6Division of Nephrology, Department of Internal Medicine, Hsin-Jen Hospital, New Taipei City 242009, Taiwan; hh258527@ms23.hinet.net

**Keywords:** NLRP3 inflammasome, indoxyl sulfate, uremia, innate immunity

## Abstract

The NLRP3 inflammasome is responsible for the maturation of caspase-1 and interleukin-1β (IL-1β). Despite the study about basal activity of the NLRP3 inflammasome in hemodialysis (HD) patients, little is known about its inducibility in the milieu of uremia. Peripheral blood mononuclear cells (PBMCs) isolated from 11 HD patients and 14 volunteers without a history of chronic kidney disease, as well as macrophages with or without the uremic toxin indoxyl sulfate (IS) pretreatment, underwent canonical NLRP3 inflammasome induction. Despite the high plasma levels of IL-1β in HD patients, caspase-1 and IL-1β in the PBMCs of HD patients remained predominantly immature and were not secreted in response to the canonical stimulus. In addition, while IS alone facilitated the inflammasome-independent secretion of IL-1β from macrophages, IS exposure before induction reduced the inducibility of the NLRP3 inflammasome, characterized by insufficient maturation of caspase-1. The low expression of inflammasome components, which was observed in both IS-pretreated cells and the PBMCs of HD patients, was probably responsible for the low inducibility.

## 1. Introduction

Patients with chronic kidney disease (CKD), particularly those on long-term hemodialysis (HD), exhibit persistent, low-grade inflammation [1]. One feature of chronic inflammation is high levels of circulating proinflammatory cytokines [2], which are produced mostly by aberrantly activated monocytes [3]. The molecular mechanism of cytokine dysregulation is still not fully understood, but emerging evidence suggests the involvement of the nucleotide binding and oligomerization domain-like receptor family pyrin domain-containing 3 (NLRP3) inflammasome.

The NLRP3 inflammasome is a cytosolic multiprotein complex composed of the intracellular sensor NLRP3, the adaptor apoptosis-associated speck-like protein containing a caspase recruitment domain (ASC), and procaspase-1. Complete assembly of the inflammasome leads to the self-cleavage of procaspase-1 to active caspase-1 that proteolytically cleaves the cytokine precursor prointerleukin-1β (pro-IL-1β) to biologically active IL-1β, which is then released into the milieu [4]. The formation of the NLRP3 inflammasome is tightly regulated and requires both priming and activation signals. The priming signal, frequently referred to as signal 1, can be provided by lipopolysaccharide (LPS) and can induce the cellular transcription of NLRP3 and pro-IL-1β. The assembly of the NLRP3 inflammasome requires a further activation signal, referred to as signal 2, which is usually provided by the bacterial toxin nigericin or other infection or stress-associated molecules [4]. The aim of inflammasome regulation is to avoid excess cytokine production and to restrict this innate immune process to only the circumstances of microbial invasion [4,5].

However, the regulation of the NLRP3 inflammasome seems to be violated in various chronic diseases, such as diabetes mellitus and CKD [4,6]. Peripheral blood mononuclear cells (PBMCs) isolated from naïve type 2 diabetic patients have aberrantly high NLRP3 inflammasome activity in response to in vitro stimulation [6]. PBMCs from HD patients also have higher transcript levels of NLRP3, ASC, caspase-1, and IL-1β than those from healthy subjects [7]. Uremic toxins may contribute to this dysregulation. For example, indoxyl sulfate (IS), one of the best-characterized protein-bound uremic toxins, is capable of inducing the release of IL-1β and other proinflammatory cytokines from macrophages with or without the aid of lipopolysaccharide (LPS) [8,9,10,11]. Whether these IS-driven reactions are NLRP3 inflammasome-dependent deserves further investigation.

Most CKD-related inflammasome dysregulation studies have focused on the basal activity but not the inducibility of the NLRP3 inflammasome in response to acute disturbance. The latter is unique because it may represent the efficacy of the inflammasome to act as a host defense system. In the present study, the inducibility of the NLRP3 inflammasome in PBMCs was compared first between HD patients and individuals without a history of CKD and then between in vitro macrophages with or without IS exposure. Our results showed that, despite the high level of circulating IL-1β in HD patients, caspase-1 and IL-1β in the PBMCs of HD patients remained immature and were not secreted in response to canonical NLRP3 inflammasome induction. Low inducibility of the NLRP3 inflammasome was also observed in macrophages exposed to IS, which might be the consequence of low IS-mediated expression of the NLRP3 machinery.

## 2. Results

A total of 24 volunteers without a history of CKD and 11 HD patients were recruited for the measurement of plasma IL-1β concentrations, while 14 non-CKD individuals and 11 HD patients underwent PBMC isolation and stimulation test. The characteristics of these patients are shown in Table 1. In comparison with the individuals without a history of CKD, the HD patients were older, predominantly male, and tended to have comorbidities of hypertension and diabetes mellitus (DM). The quantities of PBMCs isolated from the 20 mL of blood were lower in the HD patients than in the non-CKD individuals.

### 2.1. Inducibility of the NLRP3 Inflammasome in HD Patients and Non-CKD Individuals

The plasma concentrations of IL-1β were higher in HD patients than in non-CKD individuals (0.047 (interquartile range, IQR 0.022–0.069) vs. 0.091 (0.061–0.124) pg/mL, *p* = 0.027) (Figure 1a). Restricting the comparison to only those consenting to the subsequent PBMC isolation (6 HD patients and 9 non-CKD individuals) did not violate the trend (0.032 (0.019–0.065) vs. 0.100 (0.067–0.168) pg/mL, *p* = 0.012) (Figure 1b). In contrast, the baseline supernatant concentrations of IL-1β secreted from PBMCs were not higher in the HD patients than in the non-CKD individuals (13.17 (1.72–48.40) vs. 39.36 (11.89–79.45) pg/mL, *p* = 0.081) (Figure 2a). Furthermore, and to our surprise, the IL-1β secretion upon inflammasome induction was significantly lower in the HD patients than in the non-CKD individuals (15.39 (2.17–54.78) vs. 269.7 (29.0–1120.0) pg/mL, *p* = 0.020) (Figure 2a). These results indicate that the inducibility of the NLRP3 inflammasome in response to a pathogen-associated molecular pattern is attenuated in the immune cells of HD patients despite the fact that the high plasma IL-1β levels suggests a basal inflammatory status in these patients.

To confirm and explore the causes of low inducibility of the NLRP3 inflammasome in HD patients, we performed immunoblot analysis of 9 non-CKD individuals and 6 HD patients who consented to PBMC isolation. Given that the results were somewhat heterogenous (Appendix A), we recruited 5 additional HD patients and 5 additional non-CKD controls for simultaneous PBMC stimulation to ensure that the experimental conditions were identical. The overall results of the 14 non-CKD individuals and the 11 HD patients were shown in Figure 2b. While the canonical inflammasome stimulus successfully induced the secretion of caspase-1 and IL-1β from the PBMCs of non-CKD individuals, the response of PBMCs from HD patients was minimal and nonsignificant (Figure 2b). In the cell lysates, the expression levels of ASC, procaspase-1, and pro-IL-1β were lower for HD patients than for non-CKD individuals, a pattern that was consistent in both the pre- (before induction) and post-induction states (Figure 2b). Intriguingly, ASC and procaspase-1 expression was also suppressed by the induction per se (Figure 2b). Taken together, these results suggest that HD patients have a less induction-responsive NLRP3 inflammasome in the circulation than those without CKD, which might be secondary to the insufficient supply of essential inflammasome components, including the adaptor ASC and the key enzyme caspase-1.

### 2.2. The Response to Priming Stimulus in HD Patients and in Non-CKD Individuals

The canonical induction of the NLRP3 inflammasome requires priming and activation signals. PBMCs from 9 of the non-CKD individuals and 6 of the HD patients were subjected to LPS priming. LPS priming alone failed to induce significant pro-IL-1β expression in the PBMCs derived from the HD patients (Figure 3). This finding might explain why there was no elevation of IL-1β in the supernatant after LPS treatment in the HD group, whereas LPS could induce IL-1β production in the non-CKD group. The expression of the other NLRP3 inflammasome components, including caspase-1, did not differ significantly between HD patients and non-CKD individuals. These results suggest that PBMCs of HD patients are resistant to the canonical NLRP3 priming signal.

### 2.3. Effects of IS Exposure on the Inducibility of the NLRP3 Inflammasome

We postulate that uremic toxins may contribute to the poor inducibility of the NLRP3 inflammasome in the PBMCs of HD patients. To test this hypothesis, macrophages differentiated from THP-1 cells were exposed to IS, a representative uremic toxin, prior to combination treatment with LPS and nigericin. IS treatment alone downregulated the expression of NLRP3 protein but induced the secretion of IL-1β (Appendix A). By contrast, IS treatment prior to inflammasome induction did not enhance the supernatant level of IL-1β but suppressed the maturation of caspase-1 (Figure 4). The post-induction protein levels of NLRP3 were also lower in the IS-exposure group than in the control group (Figure 4). Lowering LPS priming concentration to 100 ng/mL did not alter the primary results of caspase-1 maturation being suppressed by IS pretreatment (Appendix A). These results suggest that exposure of macrophages to IS results in insufficient supplies of the inflammasome machinery, leading to a low yield rate of caspase-1 upon inflammasome induction.

We further analyzed the effects of IS exposure on LPS priming. Solitary LPS treatment resulted in borderline IL-1β secretion (Figure 5) but no caspase-1 maturation (data not shown). Compared to LPS priming alone, IS exposure prior to LPS priming abolished the induced expression of NLRP3 protein (Figure 5). We also tried low concentration (100 ng/mL) of LPS, but it did not trigger significant priming reactions such as NLRP3 or pro-IL-1β expressions (Appendix A). Therefore, at least with respect to the NLRP3 protein expression, the NLRP3 priming signal is disturbed by IS.

## 3. Discussion

This study is the first to demonstrate how immune cells isolated from the blood of HD patients respond to canonical NLRP3 inflammasome induction signals. We found that both the expression and the maturation of the NLRP3 inflammasome were suppressed despite that the immune cells were isolated from a milieu of chronic inflammation. The inflammasome pathway was perturbed both in the priming step, with low expression levels of pro-IL-1β, and in the assembly and activation steps due to the lack of the adaptor protein ASC and the effector protein caspase-1. The uremic toxin IS might mediate the low inducibility of the inflammasome, as IS exposure prior to canonical inflammasome induction diminished the expression of NLRP3 and the maturation of caspase-1, consistent with the pattern observed in the PBMCs of HD patients.

The reduced trigger potential of the NLRP3 inflammasome in HD patients is of clinical importance because the NLRP3 inflammasome is indispensable for the immune response to pathogenic intruders. The NLRP3 inflammasome is activated during various infections, and animals with NLRP3 deficiency have a high mortality rate upon infection [5]. ASC also contributes to host defense through either inflammasome-dependent or -independent mechanisms [12,13]. Clinically, infection is among the leading causes of mortality in HD patients [14], but the reason for the increased infection risk in these patients is not fully understood. Our study provides a plausible explanation that the immunity of HD patients may be compromised because of the poor inducibility of the NLRP3 inflammasome. This hypothesis is further strengthened by the use of the bacterial toxins, LPS and nigericin, as stimuli in this study.

The results of our study are different from, but not opposite to, those of a study conducted by Granata et al., in which the authors concluded that the basal NLRP3 inflammasome activity was elevated in the PBMCs of HD patients [7]. Several limitations of their study deserved attention. First, in terms of determining the NLRP3 inflammasome activity, the transcript levels, though elevated, do not outweigh the importance of protein levels. Second, only the intracellular protein levels and not the secretion of caspase-1 and IL-1β were assessed by Granata et al., and the latter, measured in our study, was more representative of the inflammasome activity. Third, the authors normalized the intracellular caspase-1 and IL-1β expression to that of the corresponding procaspase-1 and pro-IL-1β instead of the expression of the housekeeping β-actin. However, these precursor proteins tended to decrease in the immune cells of HD patients, as shown in our study and in the immunoblot images (though not quantified) presented by the authors themselves. Therefore, one cannot be sure that the total yield of active caspase-1 and IL-1β increased in their study. Finally, Granata et al. treated some PBMCs from the HD patients with canonical inflammasome stimuli, but unlike in our study, the responses were not compared with those of the PBMCs from the healthy subjects. Taken together, the evidence provided by Granata et al. was not strong enough to demolish our conclusion that both the basal machinery supply and the inducibility of the NLRP3 inflammasome were reduced in the immune cells of HD patients.

Our study attributes the low inducibility of NLRP3 inflammasome to IS, a protein-bound, poorly dialyzable uremic toxin. IS promotes the release of pro-inflammatory cytokines, including IL-1β, through various pathways, but the NLRP3 inflammasome is not one of them [8,10,15]. On the contrary, NLRP3 and caspase-1 expression were low after IS exposure [10,15]. Our results are not only compatible with previous findings but also suggest that the IS-mediated downregulation of the inflammasome machinery indeed disturbs the immune response to pathogen-associated molecules, at least in regard to the maturation of caspase-1 and IL-1β. Intriguingly, since IS alone induced IL-1β without activating caspase-1 in our study (Appendix A) and others [15], IS seems to downregulate NLRP3 via pathways unrelated to inflammasome induction and consumption. Other CKD-related factors, such as vitamin D3 deficiency and hyperphosphatemia, may also disturb the functions of the inflammasome, as vitamin D3 is crucial for NLRP3-dependent IL-1β secretion and hyperphosphatemia polarizes macrophages into the M2-like phenotype [16,17,18]. In addition, CKD manifest as a state of persistent, low-grade inflammation [1], which desensitizes immune cells during an actual infection. For example, prolonged LPS treatment alters the composition of NF-κB dimers and downregulates NLRP3 expression [19,20]. The reduced intracellular level of ASC in our HD patients, but not in the IS-treated macrophages, could also be a result of chronic inflammation, as the ASC specks and the NLRP3 inflammasome might leak extracellularly [21,22]. Taken together, since our results and mounting evidence suggest a suppressed NLRP3 inflammasome response in CKD patients, further studies are warranted to search for the underlying mechanisms.

Caution is required due to the limitations of our study. First, it remains undefined whether the characteristics of our HD patients, such as older age, male predominance, and multiple comorbidities, contributed to the attenuated inflammasome induction. However, these features are common in HD patients; hence, our findings might still be representative of the HD population. Second, the IS concentration 1 mM (213 mg/L) in our study is nearly the maximum instead of the mean circulating level in uremic patients [23]. However, 1 mM IS does not impair the viability but stimulates the maximal inflammatory reaction of THP-1 cells [10], and the inflammatory reactions are accelerated not only by IS but also by various uremic toxins in vivo. Finally, the PBMCs from our HD patients might react differently in vivo and in vitro. Nevertheless, the in vitro effect is supposed to be minimal since the cells were treated immediately after isolation, and the results of our in vitro study were compatible with those obtained using PBMCs.

## 4. Conclusions

This study is the first to demonstrate that the inducibility of the NLRP3 inflammasome in response to bacterial toxins is diminished in immune cells derived from CKD patients, which might be due to exposure to uremic toxins. Our study proposes a plausible mechanism for the high infection risk observed in the CKD population. Further studies to identify CKD-specific factors responsible for the immune suppression effects are warranted.

## 5. Materials and Methods

### 5.1. Hemodialysis Patients and Volunteers without CKD

Patients over 20 years old and on regular HD at E-DA Hospital for more than 3 months were enrolled. For comparison, volunteers over 20 years old without a history of CKD were recruited via a public announcement at E-DA Hospital. Subjects with a history of human immunodeficiency virus infection, malignancy, or current use of sulfonylurea were excluded. The baseline characteristics of the HD patients and non-CKD volunteers, including age, sex, past histories of diabetes mellitus (DM) and hypertension, current medications (in particular the use of antidiabetic agents) and the duration of HD, were recorded at the time of enrollment. The study was conducted according to the study protocols approved by the Institutional Review Board of the E-DA Hospital (permit number: EMRP-102-046 and EMRP-105-057, the approval date: 23 October 2013 and 27 October 2016, respectively).

A total of 24 volunteers without a history of CKD and 11 HD patients were recruited for the measurement of plasma IL-1β concentrations. Among them, 9 non-CKD individuals and 6 HD patients consented to and successfully completed the PBMC stimulation test. Each of these PBMC samples was divided into the following 3 groups: control, LPS treatment alone, and LPS plus nigericin treatment. However, PBMC isolation and stimulation tests for the non-CKD individuals and for the HD patients were not performed on the same day. To ensure that the difference in HD and non-CKD samples was not merely due to different experimental conditions, we further recruited 5 additional HD patients and 5 matched controls without CKD, directly testing their NLRP3 inflammasome inducibility simultaneously. These additional PBMCs were tested for only the control versus LPS plus nigericin group and not for the LPS solitary treatment group.

### 5.2. PBMC Isolation and NLRP3 Inflammasome Induction

A 20 mL aliquot of heparinized whole blood was obtained from each subject for PBMC isolation. The isolation of PBMCs was performed using Ficoll-Paque (GE Healthcare) gradient separation. In detail, the whole blood was centrifuged at 1000× *g* and 4 °C for 10 min. The plasma was removed, and the same volume of 0.1% BSA-PBS was added to resuspend the remaining cells. A total of 7 mL of Ficoll-Paque was placed into a 15 mL Falcon tube, and 7 mL of diluted blood was carefully added on top of the Ficoll-Paque. The Falcon tube was centrifuged at 1200× *g* at room temperature for 20 min. After centrifugation, the PBMC layer was carefully removed to a new 50 mL Falcon tube and resuspended in 0.1% BSA-PBS to a final volume of 40 mL. The tube was centrifuged at 1000× *g* and 4 °C for 10 min. The supernatant was removed, and the cell pellet was resuspended in 40 mL of 0.1% BSA-PBS for another centrifugation at 1000× *g* and 4 °C for 10 min. After the last centrifugation, the supernatant was removed, and the cell pellet was resuspended in 5 mL of 0.1% BSA-PBS. The viability and number of the isolated PBMCs were checked, and then the PBMCs were cultured in 2% FBS-RPMI-1640 (FCS, GIBCO, USA) for experiments. For induction of the NLRP3 inflammasome, PBMCs were treated with 0.5 μg/mL LPS for 3.5 h, followed by 1 μg/mL nigericin for 30 min. PBMCs treated with 2% FBS-RPMI-1640 but not LPS and nigericin were used as the controls. After treatment, the cell-free culture medium and the whole cell lysates were collected separately.

### 5.3. In Vitro Macrophage Treatment

THP-1 human monocytic leukemic cells (Bioresource Collection and Research Center, Taiwan) were maintained in complete culture medium (RPMI 1640, 10% heat-inactivated fetal bovine serum [24], 1% glutamine, and 1% penicillin/streptomycin) and were incubated at 37 °C in humidified air with 5% CO_2_. THP-1 cells at a density of 1 × 10^6^/mL were differentiated into macrophages by incubation with 100 nM phorbol-myristate-acetate (PMA, Sigma) for 48 h. Thereafter, the THP-1-derived macrophages were treated with FBS-free, fresh medium alone or containing 1.0 mM IS (Sigma) for 24 h. Then, the IS was removed, and the cells were treated with 0.5 μg/mL LPS for 3.5 h with or without subsequent treatment with 1 μg/mL nigericin for 30 min.

### 5.4. ELISA and Western Blot Analysis

The concentration of IL-1β in the plasma of subjects and in the culture medium of PBMCs was determined by enzyme-linked immunosorbent assay (ELISA). The protein levels of IL-1β and caspase-1 in the culture medium, as well as the protein levels of NLRP3, ASC, procaspase-1, and pro-IL-1β in the cell lysates, were determined by Western blot analysis. Briefly, each well of an SDS-PAGE gel was loaded with equal amounts of protein for electrophoresis. The resolved proteins were then electroblotted onto PVDF membranes. The membranes were blocked with 5% fat-free milk or 3% BSA and exposed to primary antibodies overnight at 4 °C. After intensive washing, the blots were incubated with secondary antibodies. The antigens on the blots were revealed with the aid of an enhanced chemiluminescence (ECL) kit from GE Healthcare.

### 5.5. Statistics

Statistical analyses were conducted with JMP statistical software (SAS Institute). The measured values were presented as the median (IQR) unless otherwise specified. The human PBMC data were analyzed using non-parametric statistics because of the skewed distribution, with Mann–Whitney U test for independent samples such as the protein levels of HD patients versus non-CKD volunteers, and Wilcoxon signed rank test for paired samples such as PBMCs derived from the same patients with versus without in vitro treatment. The independent Student’s *t*-test was used to compare the results of THP-1-dervied macrophages with versus without IS pretreatment. The difference was considered statistically significant at values of *p* < 0.05.

## Figures and Tables

**Figure 1 toxins-13-00038-f001:**
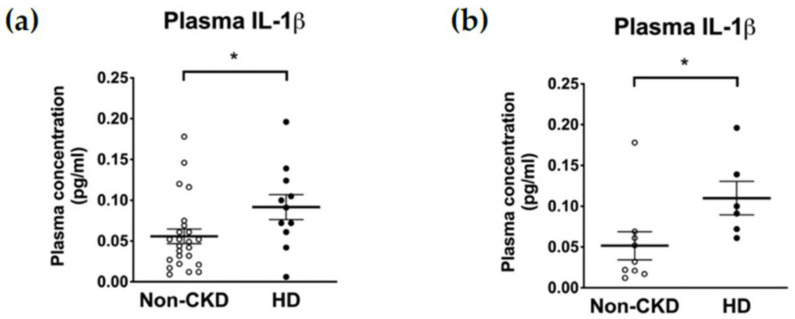
The plasma levels of IL-1β determined by ELISA. (**a**) Results of the whole population including 24 non-CKD individuals and 11 HD patients. (**b**) Results of the 9 non-CKD individuals and the 6 HD patients who consented to the subsequent immunoblot analysis. * *p* ≤ 0.05.

**Figure 2 toxins-13-00038-f002:**
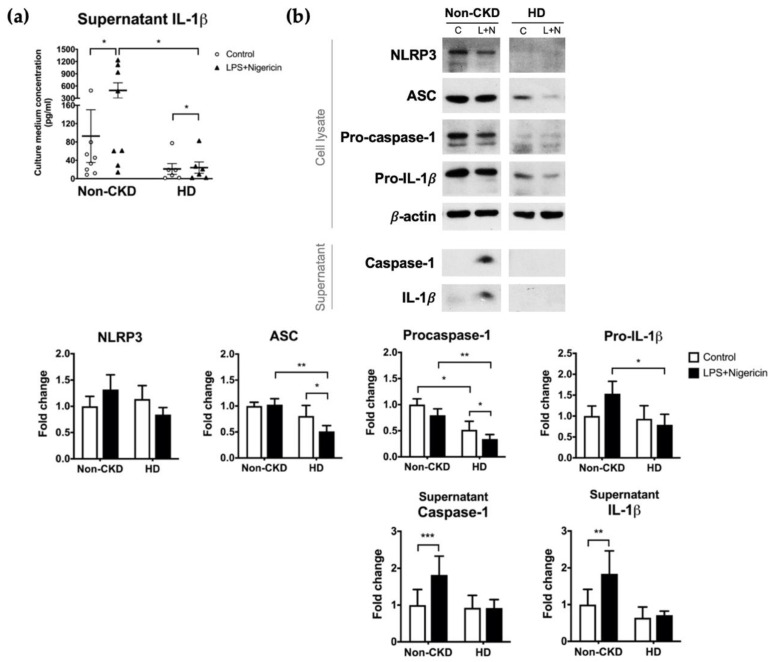
PBMC responses to NLRP3 inflammasome induction. NLRP3 inflammasome was induced by the combination of LPS and nigericin treatment. (**a**) The supernatant concentration of IL-1β in 9 non-CKD individuals and for 6 HD patients as determined by ELISA. (**b**) Immunoblot analysis of the cell lysates and supernatants of PBMCs isolated from 14 non-CKD individuals and from 11 HD patients. The figures show representative images and quantification of the immunoblot analyses, quantified relative to the non-CKD individuals with control treatment. The results are presented as a bar graph and expressed as the mean ± SEM. The results for the supernatants are corrected according to the cell number. C, control treatment; L+N, LPS plus nigericin treatment. * *p* ≤ 0.05, ** *p* ≤ 0.01, *** *p* ≤ 0.001.

**Figure 3 toxins-13-00038-f003:**
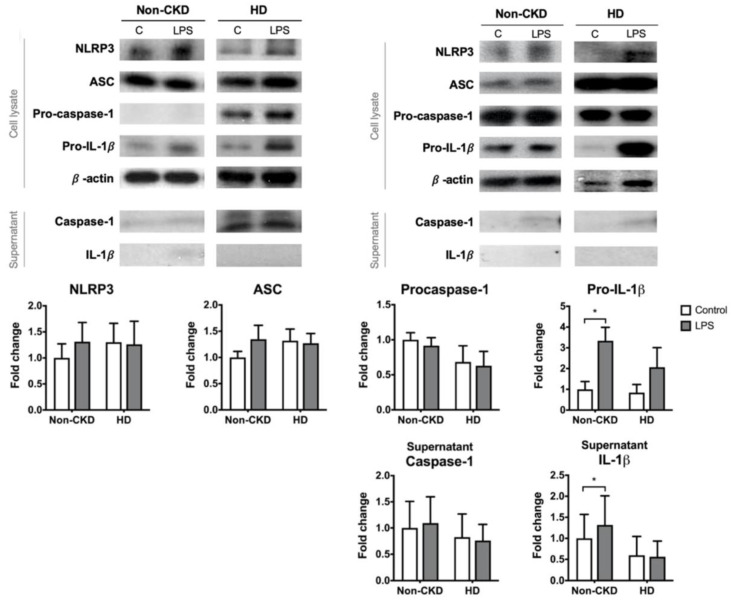
The PBMC responses to the priming stimulus. PBMCs were treated with control medium or primed with LPS. The figures show representative images and quantification of the immunoblot analyses of the cell lysates and supernatant of PBMCs isolated from 9 non-CKD individuals and 6 HD patients. * *p* ≤ 0.05.

**Figure 4 toxins-13-00038-f004:**
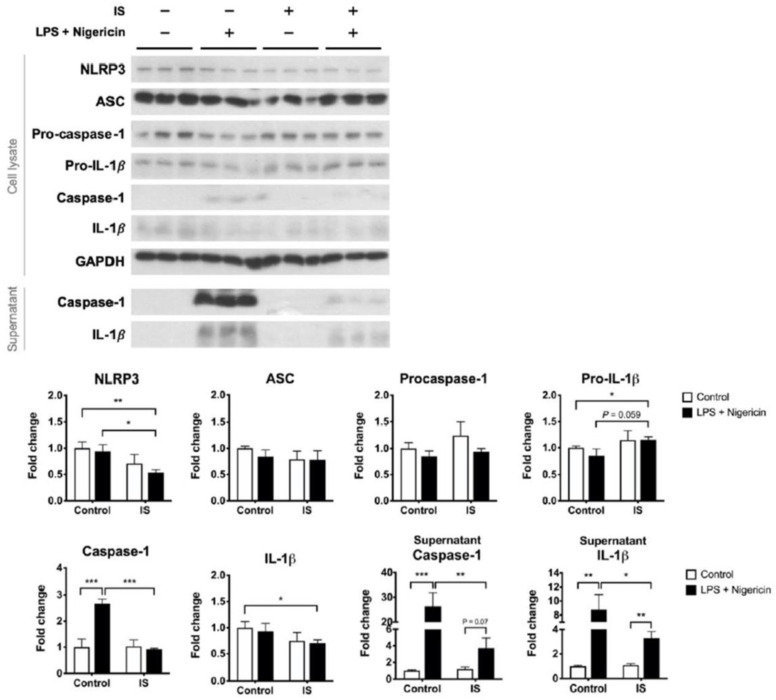
The effects of indoxyl sulfate (IS) on the inducibility of the NLRP3 inflammasome. THP-1-derived macrophages were treated with control medium or IS for 24 h and then treated with 0.5 μg/mL LPS plus 1 μg/mL nigericin for NLRP3 inflammasome induction, *n* = 6 for each group. Sup, supernatant. * *p* ≤ 0.05, ** *p* ≤ 0.01, *** *p* ≤ 0.001.

**Figure 5 toxins-13-00038-f005:**
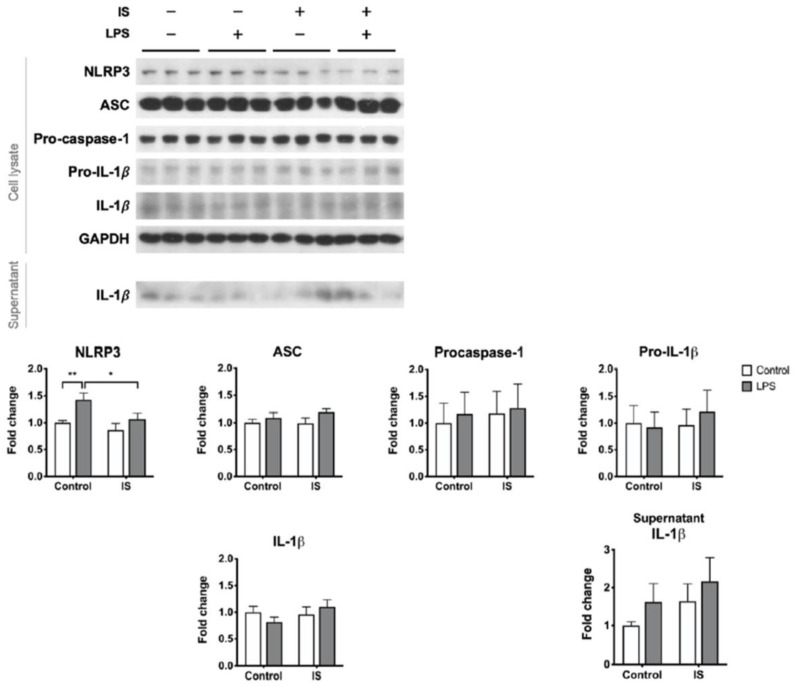
The effects of indoxyl sulfate (IS) on the responses to the priming stimulus. THP-1-derived macrophages were treated with control medium or primed with 0.5 μg/mL LPS with or without 24 h IS pretreatment. *n* = 9 for each group. * *p* ≤ 0.05, ** *p* ≤ 0.01.

**Table 1 toxins-13-00038-t001:** Subject characteristics.

	ELISA for Plasma IL-1β		PBMC Stimulation	
	Non-CKD (*n* = 24)	HD (*n* = 11)	Non-CKD (*n* = 14)	HD (*n* = 11)
Age, years (median (IQR))	57 (46–61)	59 (53–70)	61 (45.5–65.5)	64 (58–70)
Sex, male/female	11/13	8/3	6/8	8/3
HD duration, months (median (IQR))		43 (26–88)		33 (25–57)
Hypertension, *n* (%)	6 (25%)	9 (82%)	5 (36%)	9 (82%)
Diabetes mellitus, *n* (%)	3 (13%)	4 (36%)	1 (7%)	6 (55%)
PBMC number, *n* × 105			9.7 ± 13.0	6.1 ± 2.7

## Data Availability

Data are available upon request, please contact the contributing authors.

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
