# Peer review of "Indoxyl Sulfate Mediates the Low Inducibility of the NLRP3 Inflammasome in Hemodialysis Patients"

_toxins, 2021, doi:10.3390/toxins13010038_

Round 1

Reviewer 1 Report

The authors investigated the NLRP3 inflammasome inducibility in the milieu of uremia.  

Despite the well-known high basal activity of the NLRP3 inflammasome in hemodialysis  (HD) patients, the NLRP3 inflammasome of HD-PBMCs seemed not to be activated since the response of PBMCs from HD patients to canonical stimulus was nonsignificant compare to non-CKD-PBMCs (Figure 1 and 2).

The authors reported that high plasma IL-1beta levels may suggest high basal inflammasome activity in HD patients, while, the basal inflammasome activity in the PBMCs of HD patients seemed to be low (from the low levels of IL-1beta)  as showed by Figure 1, which is followed by a low inflammasome activation (low levels of IL-1beta compared to simulated non-CKD PBMCs).

Further, from WB image (Figure 2b), NLRP3 seemed higher in C than L+N (in non CKD), as for HD . In the bar graph NLRP3 was described lower in C than L+N: Please explain the apparent incongruity. More, from WB image (Figure 2b), NLRP3 seemed higher in non-CKD C compared to HD C. At the contrary, the bar graph showed no differences between C in the two groups (non-CKD and HD), while for the other variables images and bar graph were congruent. The authors should report the single data, i. e. as box plots.

Based on these data (Figur 2b) it seems that in the HD-PBMCs the inflammasome is already less active at basal compared to basal non-CKD-PBMCs. Could this influence the inductivity of inflammasome? The authors should well discuss this issue to assess the validity of their data.

Reviewer 2 Report

The reviewer agrees with the general conclusions however not with the presentations of those conclusions. Most of the studied parameters show a very skewed distribution and should be presented as medians and IQRs and handled with non-parametric statistics. 

Round 2

Reviewer 1 Report

Dear Authors

I thank you for the clearest explanation to my comments. However, it must be admitted that the heterogeneity of the results shown in Figure 2b may confuse the readers. Supplemental Figure 1 does not help to clarify these results. A method of normalizing the OD belonging to the different WB membranes should contribute to obtain bar graphs more appropriate to the figures (Figure 2b). This is my last suggestion to make the work, complex in itself, clearer to future readers.

This manuscript is a resubmission of an earlier submission. The following is a list of the peer review reports and author responses from that submission.

Round 1

Reviewer 1 Report

Comments to Toxins-882044

The authors investigated the NLRP3 inflammasome inducibility in the milieu of uremia.  

Comments to Results

Despite the well-known high basal activity of the NLRP3 inflammasome in hemodialysis  (HD) patients, the NLRP3 inflammasome of HD-PBMCs seemed not to be activated since the response of PBMCs from HD patients to canonical stimulus was nonsignificant compare to non-CKD-PBMCs (Figure 2a).

The authors reported that high plasma IL-1beta levels may suggest high basal inflammasome activity in HD patients, while, the basal inflammasome activity in the PBMCs of HD patients seemed to be low (from the low levels of IL-1beta)  as showed by Figure 1a, which is followed by a low inflammasome activation (low levels of IL-1beta compared to simulated non-CKD PBMCs): The author have to clarify and discuss this.

Further, from WB image (Figure 2b), NLRP3 seemed higher in C than L+N (in non CKD), as for HD . In the bar graph NLRP3 was described lower in C than L+N: Please explain the apparent incongruity. More, from WB image (Figure 2b), NLRP3 seemed lower in non-CKD C compared to HD C. At the contrary, the bar graph showed no differences between C in the two groups (non-CKD and HD), while for the other variables images and bar graph were congruent. The authors should report the single data, i. e. as box plots.

Based on these data (Figure 1a and 2b) it seems that in the HD-PBMCs the inflammasome is already less active at basal compared to basal non-CKD-PBMCs. Could this influence the inductivity of inflammasome? The authors should well discuss this issue to assess the validity of their data.

Figure 3. Why the WB images were not reported? Please explain.

Reviewer 2 Report

In this article, authors show that canonical stimulation of the NLRP3 inflammasome is not well induce in HD patients. This mechanism could explain the high incidence of infection in HD patients.

One of the main markers of NLRP3 activation is the secretion of IL1 B (and Il18). The authors use 2 models: PBMCs from HD patients and THP1 cells treated with IS.

For PBMC, the major problem is that they did not confirm the secretion of IL1B in PBMC when NLRP3 was not canonical stimulated (as descrided in litterature)

For THP1 treated with IS and LPS, they concluded that IS "exert little or no effect". In their experiment, IL1B secretion seems increased in TPH1 treated with LPS and IS vs LPS alone but p is 0,076 (almost significant). they conducted their study in n=3 (fig 5a). Indeed, other experiments need to be carried out (n=5 for example). In the same way, the secretion of IL18 could be of interest. Do the authors have this data and could they give us the result?

Moreover, the data presented in this study are not consistent with other studies even if previous studies cited by authors did not use the same protocol and did not study the effect of IS or HD in canonical activation.

In this study they did not demonstrated the increase in the supernatant of IL1B in HD PBMCs without induction as previously mentioned in Granata et al, Plos one (DOI:10.1371/journal.pone.0122272). Furthemore, in Granata study, HD PBMCs treated with LPSs show caspase 1 production ans IL1B and IL18 induction (the authors indicate that in Granata et al, NLRP3 was not induce)

In the same vew, Wakamatsu et al, Toxins 2018 (doi:10.3390/toxins10030124) reported that IS induced proIL1B production in cell lysate. Again, the current study does not confirm this result (Fig 4a).

Data that disagree should be better mentioned in the discussion.

I note some problems in protocol and in the in vitro experiments

First of all, the authors should have presented patients without sulfonylurea use and should have excluded HD patients treated with a sulfonylurea in all their experiments due to a confounding bias.

In the experiments using PBMCs, there is no mention of the number of patients/controls used and how they were selected. (fig 1a 23 non IRC and 20 HD controls, fig1b 8 non IRC and 10 HD, fig 2 18 non IRC and 11 HD, fig3 13 non IRC and 6 HD). Similarly, cells from patients taking a sulfonylurea are not mentioned in all experiments.

In figure 1b, the authors concluded that IL1B is secreted by non-CKD PBMCs during the induction of inflammasome. However, in Figure 1B it can be seen that half of the non-CKD CMBPs did not increase IL1B after induction. How can the authors explain this disparity?

In figure 2a, for HD, a positive immunoblot control is required for the caspase1 and IL1B experiments because nothing is visible in the blot ( HD ) when actin B is present in the cell lysates experiments.

In figure 2b, results with caspase1 were not presented whereas the expression of capsase 1 was presented with THP1 cells. It would be interesting to have the same data in different cell types, especially because procaspase1 is decreased in PBMC (with or without stimulation). This result is important because Granata et al reported an increase in the production of caspase 1 in PBMC of HD patients.

Fig 4. Results should be presented with control, control +induction, IS and IS + induction in the same histogram in order to compare the effect of induction in control and IS and not only the effect of IS in inducted cells.

Fig 5a: We need further experiments to test the differences between LPS alone and LPS and IS

Fig 5b, the authors should have presented their results with control (with no induction) to compare induction with LPS vs. IS+LPS.

Moreover, in all the LPS induction experiments, LPS concentration was 0.5µg/ml. Are the results similar if a more consensual concentration (100ng/ml) is used?